# Performance Comparison of Four Hepatitis E Antibodies Detection Methods

**DOI:** 10.3390/microorganisms12091875

**Published:** 2024-09-11

**Authors:** Milagros Muñoz-Chimeno, Nazaret Díaz-Sánchez, Lucía Morago, Vanessa Rodríguez-Paredes, Silvia Barturen, Álvaro Rodríguez-Recio, Maira Alejandra García-Lugo, Maria Isabel Zamora, María Mateo, Mónica Sánchez-Martínez, Ana Avellón

**Affiliations:** 1Hepatitis Unit, National Center of Microbiology, Instituto de Salud Carlos III, 28220 Madrid, Spain; 2Servicio de Microbiología, Hospital Central de la Defensa, 28047 Madrid, Spain; 3Centro de Investigación Biomédica en Red de Epidemiología y Salud Pública (CIBERESP), 28029 Madrid, Spain

**Keywords:** hepatitis E virus, HEV, chemiluminescence, antibody, IgM, IgG, anti-HEV

## Abstract

HEV antibody detection constitutes the main screening test for HEV infection. The aim of this study is to compare the sensitivity and specificity of four techniques: LIAISON^®^ MUREX DiaSorin anti-HEV IgG and anti-HEV IgM assays, Hepatitis E VIRCLIA^®^ IgM and IgG monotests, WANTAI HEV-IgM and IgG ELISA and VIDAS^®^ anti-HEV IgM and IgG tests in five panels of samples configurated according to the immunoblot (RecomLine, Mikrogen, Neuss, Germany). Anti-HEV IgM sensitivity in the acute phase was 100% in all techniques, while sensitivity, including the immediate convalescence phase, was 96.74% for LIAISON^®^, 83.14% for VIRCLIA^®^, 84.78% for WANTAI and 88.04% for VIDAS^®^. Anti-HEV IgM specificity was 100% for both LIAISON^®^ and VIRCLIA^®^. Anti-HEV IgM WANTAI agreed with VIRCLIA^®^ with a good Kappa coefficient (κ = 0.71). Anti-HEV IgG post-infection sensitivity was 100% for LIAISON^®^, VIDAS^®^ and VIRCLIA^®^ and 99% for WANTAI. Anti-HEV IgG specificity reached 97.17% for LIAISON and 88.68% for VIRCLIA^®^. Our results demonstrated a better capacity of LIAISON^®^ MUREX anti-HEV IgM than that of competitors for detecting acute infections as well as accurate anti-HEV IgG results and in how to resolve them.

## 1. Introduction

Hepatitis E is a liver disease caused by the hepatitis E virus (HEV). The HEV is a non-enveloped, single-stranded positive-sense RNA virus belonging to the *Hepeviridae* family, which has two subfamilies, *Parahepevirinae* and *Orthohepevirinae* [1]. Eight genotypes (HEV-1 to HEV-8) within the species *Paslahepevirus balayani* (subfamily *Orthohepevirinae*) have been described as infecting humans and a considerable number of animal species. While HEV-1 and HEV-2 exclusively infect humans, HEV-3 and HEV-4 infect humans and several other animals, mainly pigs [2,3]. Based on the recent data coming from seroprevalence and blood donors’ studies, it is estimated that there are at least two million of locally acquired HEV genotype 3 infections in Europe every year [4,5,6]. In fact, the World Health Organization (WHO) stated precise objectives regarding HEVs, including the importance of complete solutions for diagnosis and for seroprevalence studies [7].

The incubation period following the exposure to HEVs ranges approximately from 15 to 60 days [8]. Around three weeks post-infection, HEV RNA is detected in blood and stool, with a viremia lasting approximately three to six weeks (acute phase). At clinical onset, biochemical markers become elevated, and antibodies start rising, first immunoglobulin M (IgM) antibodies, followed right after by IgG ones (convalescence phase). The IgM antibodies are relatively short-lived, declining rapidly along the convalescence phase. However, the IgG response is early and long lasting with increasing antibody avidity over time [9].

Although the infection is usually self-limiting and resolves within 2–6 weeks, a serious disease with acute or chronic liver failure can occur in immunocompromised patients and a proportion of people with this disease can die [10]. In immunocompromised people, such as solid organ and stem cell transplant recipients, people infected by HIVs and patients with cancer and rheumatic diseases, chronic HEV infection can be difficult to diagnose due to the low anti-HEV antibody titers, and definitive diagnosis can be delayed [8]. Quick diagnosis is needed in chronic HEV infection, as liver cirrhosis and hepatocellular carcinoma apparition occurs relatively soon after infection, compared to other hepatotropic oncoviruses such as hepatitis B viruses (HBVs) or hepatitis C viruses (HCVs) [11,12]. Chronic HEV infection may induce post-transplant hepatitis, cirrhosis, liver failure and even retransplantation, which may cause a relapse of HEV infection in the newly transplanted liver [13], a situation that also needs prompt diagnosis. Both in immunocompromised and immunocompetent patients, diagnosing extrahepatic disorders can be challenging; among them, neurological and renal manifestations are the most commonly reported, which include neuralgic amyotrophy, Guillain–Barré syndrome, acute kidney injury and glomerular disease [14,15,16].

Laboratory diagnosis plays a central role in the detection of acute HEV infections and provides information on the spread of HEVs [17]. The European Association for the Study of the Liver guidelines recommend a combination of specific antibodies and viral genome detection for HEV diagnosis [8]. While HEV RNA can be detected very early, anti-HEV IgM and anti-HEV IgG antibodies provide information on acute and convalescent infections, respectively [17]. Since the specificity of certain assays is not optimal for anti-HEV IgM [18,19] and since anti-HEV IgG is often concomitant with the subacute phase, the serological testing for HEV diagnosis ideally combines both anti-HEV IgM and IgG detection [19]. Studies investigating seroprevalence and the sub-optimal performance of certain anti-HEV IgG assays with the lack of sensitivity have previously resulted in very significant underestimations of populations’ exposure to HEVs [20].

On the other hand, the knowledge of the HEV genotypes is important not only for epidemiological purposes and for disease progression but also for the development and appropriate selection of diagnostic tools [21]. Several studies indicate that some commercially available test sensitivities for certain genotypes are higher, since different antigens of various genotypes are used in different assays [17]. However, only one HEV serotype has been identified [22]. Apart from that, cross reactivity in anti-HEV IgM assays has been commonly reported against cytomegalovirus and Epstein–Barr virus antibodies (anti-CMV and anti-EBV), respectively [19].

Currently, the serological technique considered the reference standard for confirmation is the immunoblot [23] and, the ones considered having the best performance are Wantai HEV IgM and IgG ELISA [24,25,26]. Fully automated assays are now available, which include Vidas^®^ (BioMérieux, Marcy-l’Étoile, France), Virclia^®^ (Vircell, Granada, Spain) and Liaison^®^ (DiaSorin Spa, Saluggia, Italy) [27,28,29,30].

Here we evaluate the sensitivity and specificity of a marketed tool for Hepatitis E antibody detection by chemiluminescence (LIAISON^®^ MUREX anti-HEV IgG and anti-HEV IgM assays, DiaSorin), comparing it with both the immunoblot reference standard techniques and other three tests: Wantai HEV IgM and IgG ELISA (Biologic Pharmacy Enterprise, Beijing, People’s Republic of China), Vidas^®^ (BioMérieux) and VirClia^®^ (Vircell) for sensitivity and VirClia^®^ (Vircell) for specificity.

## 2. Materials and Methods

### 2.1. Clinical Sample Collection and Characterization of Panels

According to the panels below, serum samples confirmed as positive for HEV antibodies were selected among those received at the laboratory for HEV diagnosis with an enough sample volume for this study (500 µL). HEV RNA was determined as follows: RNA was automatically extracted from 200 µL of each serum sample with a MagnaPure LC 2.0© System (Roche Diagnostics, Mannheim, Germany) following total nucleic acid protocol. A 171 bp fragment of the HEV ORF1 region was amplified by nested PCR as described previously by Fogeda and colleagues [19]. Anti-HEV IgG and anti-HEV IgM were confirmed by immunoblot recomLine HEV IgG/IgM (Mikrogen, Neuss, Germany), which is considered the reference method for anti-HEV. The manufacturer spotted different parts of the recombinant capsid protein of genotypes 1 and 3 and the protein derived from ORF3 on a nitrocellulose strip. This method is designed for the separated detection of IgM and IgG antibodies against HEV-1 to HEV-4.

Sample panels were as follows: panel I included 21 samples positive to anti-HEV IgM by immunoblot and positive for HEV RNA (representing the immediate acute HEV infection). Panel II included 71 samples positive to anti-HEV-IgM by immunoblot and negative to HEV RNA (characteristic of acute HEV infection in the early convalescence phase). Panel III included 100 samples positive to anti-HEV IgG, representing past HEV infection. Panel IV included 101 samples negative to anti-HEV IgM, and panel V included 106 samples negative to anti-HEV IgG (87 of the samples included in panels IV (87/101) and V (87/106) were negative for both anti-VHE IgM and anti-VHE IgG).

### 2.2. Compared Assays

All four assays were performed according to manufacturer instructions. The following explains them briefly:

LIAISON^®^ MUREX anti-HEV IgG and anti-HEV IgM assays (DiaSorin SpA, Saluggia, Italy) (from now, LIAISON) are fully automated chemiluminescence immunoassays for the quantitative determination and qualitative detection of IgG and IgM HEV antibodies, respectively. Both of these immunoluminescent assays were coated with genotypes 1 and 3 ORF antigens. The assay was performed on a LIAISON XL instrument, and results were determined as an index automatically calculated by it. Samples with an IgM index value below the threshold value (set at 1.00) were considered negative, whereas an index value above it was considered a positive result. The quantitative Liaison^®^ Murex HEV IgG measures the intensity of luminescence, which is proportional to the concentration of HEV IgG in the sample. Test values are automatically calculated by the instrument and expressed as IU/mL. The Liaison^®^ Murex HEV IgG assay has been standardized indirectly against the WHO international standard NIBSC 95/584 [31]. IgG values below the threshold value (0.3 IU/mL) were considered negative. Using 20 uL of sample, the result were obtained in 30 min.

Hepatitis E VIRCLIA^®^ IgM and IgG monotests (VirCell S.L., Granada, Spain) (from now, VIRCLIA) are indirect chemiluminescent immunoassays to test antibodies against HEVs, suitable for the VIRCLIA monotest system (VirCell, Granada, Spain). A test strip contains all reaction vessels (n = 3) and reagent containers (n = 5), including a calibrator and a negative control, required for processing. The reaction wells of the VIRCLIA were coated only with the HEV genotype 1 antigen. The strip monotest was processed on the VIRCLIA instrument, which can process 24 samples simultaneously. Semi-quantitative IgG/IgM results are presented as relative light units (RLU) index in 45 min, using 100 µL of serum.

Wantai HEV IgM and IgG ELISA (Wantai Biologic Pharmacy Enterprise, Beijing, China) (WANTAI) are enzyme-linked immunosorbent assays for the qualitative detection of IgG and IgM HEV antibodies, based in an antigen obtained from a highly conserved region of ORF2 derived only from the genotype 1 strain. These assays have a grey zone from 0.9 to 1.1 (sample absorbance value/cut-off, i.e., an absorbance value of negative control +0.16 for IgG or 0.26 for IgM), above which the assay was considered positive. A total of 10 µL of sample and 120 min to the first result were necessary.

VIDAS^®^ anti-HEV IgM and IgG tests (BioMérieux, France) (VIDAS) are sandwich enzyme immunoassays. The VIDAS anti-HEV IgM is a qualitative assay, whose test value is automatically calculated by the instrument as an index value and expressed in U/mL. Results were negative if their IgM index value was below the threshold value (set at 1.00). The quantitative VIDAS anti-HEV IgG measures the intensity of fluorescence, which is proportional to the concentration of HEV IgG in the sample. The VIDAS anti-HEV IgG assay was standardized to internal reference calibrators titrated against the WHO international standard NIBSC 95/584. Results were considered negative if the IgG was below the threshold value (0.56 U/mL). The assay can be performed on an automated VIDAS instrument using 100 µL of serum, and results are available in 40 min.

### 2.3. Statistical Analyses

Sensitivity was calculated for each test as the true positive (TP)/(TP + false negative [FN]). The confident interval (CI) was calculated at 95%. Specificity was calculated for LIAISON and VIRCLIA tests as the true negative (TN)/(TN + false positive [FP]). Indeterminate results were considered non-negative in sensitivity and specificity. The positive predictive value (PPV) was defined as TP/(TP + FP) and negative predictive value (NPV) as TN/(TN + FN).

Both Cohen’s kappa coefficient (κ) and Pearson’s (PS) coefficient were used to study the agreement of the different used techniques. Cohen’s kappa coefficient (κ) was used for qualitative results and Pearson’s correlation coefficient (PS) for index/RLU/absorbance results. These coefficients were also estimated for anti-HEV IgM panels (I and II). κ was interpreted as follows: ≤0.20 = poor; 0.21–0.40 = fair; 0.41–0.60 = moderate; 0.61–0.80 = good and 0.81–1.00 = very good. Statistics were performed with SPSS 28.0, and *p*-values were set at 0.05.

### 2.4. Phylogenetic Analysis

A total of 15 of the 21 samples of panel I (IgM and RNA positives) were sequenced for genotyping using the previously described method [32]. Phylogenetic analysis and sub-genotypes assignation were checked according to the published study by Muñoz-Chimeno et al. [32].

## 3. Results

The performance of anti-HEV IgM is summarized in Table 1. Sensitivity of anti-HEV IgM in the acute-phase samples (Panel I) was 100% in all techniques but decreased when immediate convalescence-phase samples (Panel I + II) were included, being 96.74% (89/92) in LIAISON; 83.14% (74/89) in VIRCLIA; 84.78% (78/92) in WANTAI and 88.04% (81/92) in VIDAS. The specificity of anti-HEV IgM calculated in panel IV was 100% for LIAISON and VIRCLIA, the PPV being 100% in acute and acute-plus convalescence phases for both. The NPV was 97.11% in LIAISON and 85.59% in VIRCLIA. The raw data are available in Appendix A (Panel I and II).

Phylogenetic analysis showed that 15 to 21 samples of panel I were genotype 3, one sample was sub-genotype 3c, two samples were sub-genotype 3f-A2, and the remaining 12 samples were sub-genotype 3f-A1 (Appendix A).

Regarding anti-HEV IgG tests, results of the past HEV infection sample panel (Panel III) are shown in Table 2. All evaluated techniques except WANTAI reached 100% of anti-HEV IgG sensitivity, WANTAI being 99% (99/100). Specificity of anti-HEV IgG (Panel V) was 97.16% (103/106) for LIAISON and 88.68% (94/106) for VIRCLIA. The raw data are available in Appendix A (panel III) and Appendix A (panel IV and V).

Cohen’s kappa coefficient (κ) for qualitative results and Pearson’s (PS) coefficient for index/RLU/absorbance of anti-HEV IgM of acute and convalescence phases (Table 3) were the highest between WANTAI and VIRCLIA (κ = 0.712, *p* < 0.001; PS = 0.767, *p* < 0.001). A moderate correlation was observed when comparing VIDAS-VIRCLIA (κ = 0.491, *p* < 0.001; PS = 0.685, *p* < 0.001), LIAISON-VIDAS (κ = 0.432, *p* < 0.001; PS = 0.785, *p* < 0.001) and WANTAI-VIDAS (κ = 0.425, *p* < 0.001; PS = 0.601, *p* < 0.001). Cohen’s kappa coefficient and PS coefficient for concentrations of total anti-HEV IgM results (including all tested samples, N = 190) for VIRCLIA-LIAISON was κ =0.849 (*p* = 0,000) and PS = 0.738 (*p* < 0.001), respectively.

## 4. Discussion

Serology has been considered the first screening tool for HEV diagnosis, and it is crucial to estimate the infection burden in seroprevalence studies, but it has come across some issues. Marker identification through accurate diagnostic tests still a challenge because of the lack of agreement of sensitivity and specificity in the available assays [17,26,33]. Additionally, the anti-IgM detection of acute infection requires tests with high sensitivity due to its short-lasting [9]. In the present study, we evaluated comparatively four of the most widely used marketed anti-HEV IgM and anti-HEV IgG assays (LIAISON, VIRCLIA, WANTAI and VIDAS) and compared its sensitivity and specificity between them and with the reference standard technique.

Regarding anti-HEV IgM in the acute phase, our results demonstrate an optimal sensitivity in all evaluated techniques as other authors published when comparing VIRCLIA-WANTAI [28], VIDAS-WANTAI [29] and VIRCLIA-LIAISON [34]. However, there are differences in the convalescence phase. In our study, during the immediate convalescence phase, in which anti-HEV IgM levels are expected to decline, LIAISON showed the highest sensitivity (96.74%) followed by VIDAS (88.04%), WANTAI (84.78%) and VIRCLIA (83.14%). In recent studies, a WANTAI anti-HEV IgM assay showed the highest sensitivity in post-viremic (convalescence) samples from immunocompetent and immunocompromised patients comparing to LIAISON [30] and to VIDAS [29]. Our different findings might be related to the infection phase of the samples included in each report as well as in the immune status of the patients. In our study, the samples were anonymized, and the immune status was not known but was expected to be immunocompetent. Regarding anti-HEV IgM, LIAISON and VIRCLIA presented optimal specificity [28,34]. The lack of sensitivity in the convalescence phase and/or in immunocompromised patients may result in false-negative HEV infections, particularly if anti-HEV IgG is not determined simultaneously. If acute-infection screening is conducted with anti-HEV IgM as a unique marker, a device with the highest sensitivity should be used [9]. Although in our study, cross-reactivity was not assessed, other reports described a high degree of Epstein–Barr (EBV) and cytomegalovirus (CMV) cross-reactivity with 33% and 24% of anti-HEV IgM positive samples also testing positive for anti-EBV IgM and anti-CMV IgM, respectively [19]. On the other hand, some authors determined no anti-HEV IgM VIRCLIA and LIAISON cross-reactivity with CMV and EBV infections [28,34].

In the past ten years, HEV infection has been described as an increasing health problem in Europe [4,5,6]. Since then, a lot of studies have arisen to compare the agreement of results of different anti-HEV IgM detection techniques. In 2021, VIDAS to WANTAI agreement was assessed in positive and negative samples, being this good between them (κ = 0.750) [27]. Our study, whose agreement was calculated in acute and convalescence phase, showed moderate VIDAS–ANTAI agreement (κ = 0.425), probably due to the different samples included (only recent infections). Regarding WANTAI–VIRCLIA, agreement was good (κ = 0.712) with no other reports found comparing them. In the case of VIRCLIA and LIAISON, we observed a poor agreement in the acute and convalescence phases (κ = 0.176) and good agreement overall (positive and negative samples) (κ = 0.849) similar to previously reported results (κ = 0.930) [34]. Differences in correlation results obtained by κ and PS can be explained by discrepancies in manufacturer cut-off value because it may influence assay sensitivity [35]. Additionally, the data showed that WANTAI–VIRCLIA can be interchangeably used in a qualitative assay whereas WANTAI–VIRCLIA and LIAISON–VIDAS can be exchanged in a quantitative assay (PS = 0.785 and PS = 0.767, respectively).

In the case of anti-HEV IgG, we have found an optimal anti-HEV IgG sensitivity (100%) in LIAISON, VIDAS and VIRCLIA with a slight decrease in WANTAI (99%). A previous evaluation of a VIDAS anti-HEV IgG assay showed a good clinical sensitivity in immunocompetent patients [29] as well as other reported data for the WANTAI IgG assay in HEV RNA positive patients [36]. The LIAISON anti-HEV IgG quantitative test was compared to WANTAI test using a clinical sample, and the results agreed with the limit of detection stated for LIAISON [30]. The LIAISON IgG assay had better specificity (97.17%) than VIRCLIA (88.68%), due to a certain number of false VIRCLIA anti-HEV IgG-positive results. This specificity impairing VIRCLIA anti-HEV IgG has been previously reported and was suggested to be attributed to its high sensitivity [21]. Initially, WANTAI was considered the one with the best anti-HEV IgG performance [24,25,26,37]. Afterwards, automated commercial assays have shown a better or equal sensitivity and specificity in seroprevalence studies than conventional ELISA as WANTAI [23,30,38]. The lack of sensitivity or specificity might result in seroprevalence under or overestimation, respectively. For this reason, the correct assessment of anti-HEV IgG performance in seroprevalence studies needs to be updated [39]. In fact, the first fully automated assay (LIAISON) has been used for the first time in a seroprevalence study in Brazil [38].

According to a recent Spanish surveillance study, the most common genotype in Spain is genotype 3 [32], and in our study, all sequenced samples were genotype 3. The WANTAI assay is based on pORF2 dimers derived only from the genotype 1 strain, but previous studies have shown no sensitivity problems in the detection of infections caused by the genotype 3 HEV [40,41]. The same occurs with the VIDAS^®^ assay whose ability to detect HEV antibodies in patients infected with the genotype 1, genotype 3-rabbit or genotype 4 has been tested with a few samples [29]. Although the anti-HEV assays use genotype 1 antigens, they detect antibodies directed against other genotypes. However, several polymorphisms have been found in ORF2/ORF3 genome [42], which may affect the sensitivity in some samples despite having only one serotype. In the case of LIAISON, this assay uses the genotype 1 and 3 capsid proteins as antigens, although it has been reported to be able to reliably detect infections caused by HEV genotypes 1, 2 and 4 [23,30]. All study samples were genotype 3, and this is a limitation. For this reason, it would be useful to test more samples from other genotypes (1, 2 and 4) to compare the sensitivity and specificity in all the techniques of our study.

Our results demonstrated a better capacity of LIAISON^®^ MUREX anti-HEV IgM for detecting acute with a view to resolve infections and accurate anti-HEV IgG results for being used in seroprevalence studies. Moreover, quicker batch analysis can be performed since it is the first chemiluminescence assay that can be considered a high throughput assay for HEV serology. In addition, HEV infection may be indistinguishable from other causes of viral hepatitis. As a result, the optimum would be a one-step tool for hepatitis diagnosis in a fully automated solution. This tool may allow to meet the adoption of the 2015 resolution in the “2030 Agenda for Sustainable Development” [43] and the WHO recommendation for advocating the timely diagnosis of hepatitis viruses to prevent further transmission and reduce morbidities and mortalities [44].

## 5. Conclusions

Our results demonstrated a good capacity of LIAISON^®^ MUREX anti-HEV IgM for detecting and resolving acute infections and achieving accurate anti-HEV IgG results in seroprevalence studies. Moreover, quicker batch analysis can be performed since it is the first chemiluminescence assay that can be considered a high throughput assay for HEV serology.

LIAISON anti-HEV IgG and anti-HEV IgM assays (DiaSorin Spa, Saluggia, Italy) complement complete diagnostic panel tests for hepatitis A, B, C, D and E, which is right now, part of a cost-effective solution that can increase diagnostic efficiency, optimize resources and test timing and help limit chronic medical conditions.

## Figures and Tables

**Table 1 microorganisms-12-01875-t001:** True positive (TP), false negative (FN), the positive predictive value (PPV), the negative predictive value (NPV), sensitivity and specificity of anti-HEV IgM. * Includes indeterminate results, ** three samples were insufficient. For sensitivity and specificity calculation, indeterminate results were considered non-negative.

Technique	LIAISON	VIRCLIA	WANTAI	VIDAS
TP (acute phase)	21	20 *	21	21
FN (acute phase)	0	0	0	0
TP (acute and convalescence phase)	89	74 *	78 *	81
FN (acute and convalescence phase)	3	15	14	11
FP	0	0	-	-
TN	101	101	-	-
Indeterminate results		2	2	
PPV acute phase	100	100	-	-
PPV acute and convalescence phase	100	100	-	-
NPV acute phase	97.11	85.59	-	-
Acute phase sensitivity (CI 95%)	100% (21/21)	100% (20/20)	100% (21/21)	100% (21/21)
Acute and convalescence phase sensitivity (CI 95%)	96.74% (89/92)	83.14% ** (74/89)	84.78% (78/92)	88.04% (81/92)
	(93.11–100.37)	(75.37–90.92)	(77.44–92.12)	(81.41–94.67)
Specificity	100% (101/101)	100% (101/101)	-	-

**Table 2 microorganisms-12-01875-t002:** True positive (TP), false negative (FN), the positive predictive value (PPV), the negative predictive value (NPV), sensitivity and specificity of anti-HEV IgG in the past HEV infection sample panel. * Includes indeterminate results.

Technique	LIAISON	VIRCLIA	WANTAI	VIDAS
TP (past phase)	100	100	99	100
FN (past phase)	0	0	1	0
FP	3	12 *	-	-
TN	103	94	-	-
Indeterminate results	0	2	-	-
PPV acute phase	97.08%	90.09%	-	-
NPV acute phase	100%	100%	-	-
Past phase sensitivity	100% (100/100)	100% (100/100)	99.0% (99/100)	100% (100/100)
(CI 95%)			(96.07–101.26)	
Specificity (CI 95%)	97.17% (103/106)	88.68% (94/106)	-	-
	(94.01–100.33)	(82.65–94.71)		

**Table 3 microorganisms-12-01875-t003:** The agreement of qualitative results and the correlation of concentration/index of the evaluated anti-HEV IgM techniques in the acute and convalescence phase (N = 89).

	Cohen’s Kappa Coefficient	Pearson’s Correlation Coefficient
LIAISON-WANTAI	0.066 (*p* = 0.394) (poor)	0.562 (*p* < 0.001)
LIAISON-VIDAS	0.432 (*p* < 0.001) (moderate)	0.785 (*p* < 0.001)
LIAISON-VIRCLIA	0.176 (*p* = 0.019) (poor)	0.538 (*p* < 0.001)
WANTAI-VIDAS	0.425 (*p* < 0.001) (moderate)	0.601 (*p* < 0.001)
WANTAI-VIRCLIA	0.712 (*p* < 0.001) (good)	0.767 (*p* < 0.001)
VIDAS-VIRCLIA	0.491 (*p* < 0.001) (moderate)	0.685 (*p* < 0.001)

## Data Availability

The data presented in this study are available on request from the corresponding author. The data are not publicly available due to limitations of ethical approval involving the patient data and anonymity.

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
