# Peer review of "Performance Comparison of Four Hepatitis E Antibodies Detection Methods"

_microorganisms, 2024, doi:10.3390/microorganisms12091875_

Round 1

Reviewer 1 Report

Comments and Suggestions for Authors

The authors aim to compare the sensitivity and specificity of four commercially available HEV antibodies detection kits, including LIAISON® MUREX 12

DiaSorin, Hepatitis E VIRCLIA®, WANTAI, and VIDAS®. They utilized five panels of immunoblot-configurated serum samples and demonstrated that LIAISON® has superior capacity in detecting anti-HEV IgM-positive samples. The study provides valuable insights for the development and use of serologic diagnostic tools for HEV infection. However, some points require further clarification before publication. Here are a few comments for the authors to consider:

1.    Introduction section, lines 87-93: The authors compared Liaison with immunoblot reference standard techniques and other three tests (Wantai, Vidas, and VirClia) for sensitivity, but only with VirClia for specificity. Why did the authors not compare the Liaison with all these tests for specificity?

2.    Materials and Methods section, lines 102-106: Is there any information on sub/genotype of the HEV RNA-positive serum samples? The authors mentioned that HEV genotype is important for the development and appropriate selection of diagnostic tools in lines 74-78.

3.    Materials and Methods section, lines 112-118: Is it necessary to include a panel of samples that are neither positive for anti-HEV IgM nor anti-HEV IgG as a negative control? Additionally, for improved readability, a Table detailing the classification of these samples would be beneficial.

4.    Materials and Methods section: Is ethics committee approval required for such a study involving remaining and anonymized samples in Spain?

5.    Results section: lines 183-190: Please clarify the significance of the results of Cohen’s kappa coefficient and Person’s analyses. What do these results indicate in the context of the study?

6.    Discussion section: The authors should acknowledge the limitations of this study, such as the relatively small sample size and the samples were derived from a single country/center. Additionally, sincemost HEV RNA-positive samples in this study may belong to genotype 3 HEV, the sensitivity and specificity of these kits for other HEV genotypes need further confirmation.

7.    Discussion section, lines 289-307: The discussion of artificial intelligence methods in this section appears largely irrelevant to the present study and could be significantly condensed.

8.    General Formatting: Throughout the manuscript, both “,” and “.” are used as decimal points. Please standardize them to “.”. Additionally, please check and correct any grammatical errors.

Comments on the Quality of English Language

Minor editing of English language required.

Author Response

Comment 1.    Introduction section, lines 87-93: The authors compared Liaison with immunoblot reference standard techniques and other three tests (Wantai, Vidas, and VirClia) for sensitivity, but only with VirClia for specificity. Why did the authors not compare the Liaison with all these tests for specificity?

Response 1: We decided to compare specificity in both Liaison and Virclia because these techniques are the most used available techniques in Spain. Furthermore, Virclia has recently shown problems regarding specificity and that is why this technique was chosen.

Comment 2.    Materials and Methods section, lines 102-106: Is there any information on sub/genotype of the HEV RNA-positive serum samples? The authors mentioned that HEV genotype is important for the development and appropriate selection of diagnostic tools in lines 74-78.

Response 2: The authors include now the sub-genotype results in the manuscript. Material and methods lines 173-176 and results lines 216-218.

Comment 3.    Materials and Methods section, lines 112-118: Is it necessary to include a panel of samples that are neither positive for anti-HEV IgM nor anti-HEV IgG as a negative control? Additionally, for improved readability, a Table detailing the classification of these samples would be beneficial.

Response 3: 87 of the total samples included in panels IV (87/101) and V (87/106) are negative for both IgM anti-VHE and IgG anti-VHE. The authors will include in supplementary material a table which contain all samples classification and results of the four used techniques. Furthermore, the authors include this comment in material and methods (lines 118-119).

Comment 4.    Materials and Methods section: Is ethics committee approval required for such a study involving remaining and anonymized samples in Spain?

Response 4: In our institute (ISCIII), the ethics committee approval is required even if the study involves remaining and anonymized samples.

Comment 5.    Results section: lines 183-190: Please clarify the significance of the results of Cohen’s kappa coefficient and Person’s analyses. What do these results indicate in the context of the study?

Response 5: The authors include an explanation to Cohen’s kappa and Person’s coefficient in materials and methods (lines 166-169) and it also include the significance of the results in discussion part (lines 260-265).

Comment 6.    Discussion section: The authors should acknowledge the limitations of this study, such as the relatively small sample size and the samples were derived from a single country/center. Additionally, since most HEV RNA-positive samples in this study may belong to genotype 3 HEV, the sensitivity and specificity of these kits for other HEV genotypes need further confirmation.

Response 6: The authors include in discussion the limitations of only one sub-genotype was included (lines 295-298.

Comment 7.    Discussion section, lines 289-307: The discussion of artificial intelligence methods in this section appears largely irrelevant to the present study and could be significantly condensed.

Response 7: The authors consider the reviewer comment and eliminate this information.

Comment 8.    General Formatting: Throughout the manuscript, both “,” and “.” are used as decimal points. Please standardize them to “.”. Additionally, please check and correct any grammatical errors.

Response 8: The authors standardize “.” and corrected grammatical errors in the text. Changes can be found through the text.

Reviewer 2 Report

Comments and Suggestions for Authors

Hepatitis E virus (HEV) infection induces acute infection and, sometimes, chronic liver pathology based on the immunity of the infected subjects. Although there is a specific therapeutic maneuver to contain HEV-related liver diseases, the diagnosis of HEV is of utmost importance due to its potential to induce hepatitis in epidemic situations and to cause complications such as acute or chronic liver failure. Several kits and assessment options are available to denote HEV infection by analyzing serum samples or other specimens. The article entitled “

 Performance comparison of four Hepatitis E antibody detection methods” has been accomplished to analyze the sensitivity and specificity of different available commercial kits. They used the kits of LIAISON® MUREX 12 DiaSorin anti-HEV IgG and anti-HEV IgM assays, Hepatitis E VIRCLIA® IgM and IgG monotests, WANTAI HEV-IgM and IgG ELISA and VIDAS® anti-HEV IgM and IgG tests. The novelty of this article is not so profound as similar works on the same subjects have been published in different publications:

1.      Zhang Q, Zong X, Li D, Lin J, Li L: Performance Evaluation of Different Commercial Serological Kits for Diagnosis of 394 Acute Hepatitis E Viral Infection. Pol J Microbiol 2020, 69(2):217-222.

The authors of this article have checked similar works using different kits. Interestingly, the sensitivity and specificity of assessments varied from 80% to 99% with different kits.

In addition, other investigators have also accomplished similar works to check the sensitivity and novelty of different kits used for assay HEV positivity. The authors are also aware of these situations as they have referred to these works:

1. Eichhorn A, Neumann F, Baumler C, Gutsmann I, Grobe O, Schluter F, Muller S, Krumbholz A: Assessment of the 387 Diagnostic Performance of Fully Automated Hepatitis E Virus (HEV) Antibody Tests. Diagnostics (Basel) 2024, 14(6). 388

2. Capai L, Falchi A, Charrel R: Meta-Analysis of Human IgG anti-HEV Seroprevalence in Industrialized Countries and a 389 Review of Literature. Viruses 2019, 11(1). 390

3.  Shrestha AC, Flower RL, Seed CR, Stramer SL, Faddy HM: A Comparative Study of Assay Performance of Commercial 391 Hepatitis E Virus Enzyme-Linked Immunosorbent Assay Kits in Australian Blood Donor Samples. J Blood Transfus 2016, 392 2016:9647675.

Several other publications are available regarding this.

Thus, any new article should contain novelty and new information regarding this issue. There may be several reasons for slight differences in sensitivity and specificity between different kits based on the time of collection of sera, patients’ condition, and others.

Another area of interest is to provide a response about the utility of different kits depending on work with a multicenter character. The utility and sensitivity of the kit may be variable among different populations worldwide.

Author Response

Reviewer 2

Hepatitis E virus (HEV) infection induces acute infection and, sometimes, chronic liver pathology based on the immunity of the infected subjects. Although there is a specific therapeutic maneuver to contain HEV-related liver diseases, the diagnosis of HEV is of utmost importance due to its potential to induce hepatitis in epidemic situations and to cause complications such as acute or chronic liver failure. Several kits and assessment options are available to denote HEV infection by analyzing serum samples or other specimens. The article entitled “

Performance comparison of four Hepatitis E antibody detection methods” has been accomplished to analyze the sensitivity and specificity of different available commercial kits. They used the kits of LIAISON® MUREX 12 DiaSorin anti-HEV IgG and anti-HEV IgM assays, Hepatitis E VIRCLIA® IgM and IgG monotests, WANTAI HEV-IgM and IgG ELISA and VIDAS® anti-HEV IgM and IgG tests. The novelty of this article is not so profound as similar works on the same subjects have been published in different publications:

Comment 1.      Zhang Q, Zong X, Li D, Lin J, Li L: Performance Evaluation of Different Commercial Serological Kits for Diagnosis of 394 Acute Hepatitis E Viral Infection. Pol J Microbiol 2020, 69(2):217-222.

The authors of this article have checked similar works using different kits. Interestingly, the sensitivity and specificity of assessments varied from 80% to 99% with different kits.

Response 1: The authors of this article use four techniques but only WANTAI is the common with our study. The other techniques checked was commercial Chinese available test and it were less common in Europe. Furthermore, the authors only include 86 diagnosed cases of acute hepatitis E (positive samples) and 96 healthy volunteers (negative samples).  The rest of the samples was to check cross-reactions. Therefore, the number of samples is less than our study and the techniques used change.

Comment 2: In addition, other investigators have also accomplished similar works to check the sensitivity and novelty of different kits used for assay HEV positivity. The authors are also aware of these situations as they have referred to these works:

  1. Eichhorn A, Neumann F, Baumler C, Gutsmann I, Grobe O, Schluter F, Muller S, Krumbholz A: Assessment of the 387 Diagnostic Performance of Fully Automated Hepatitis E Virus (HEV) Antibody Tests. Diagnostics (Basel) 2024, 14(6). 388

Response 2: The authors in this article only use 53 samples to compare HEV-IgM (Table S2) and not tested convalescence phase. In the case of HEV-IgG panel, they use 100 samples but only MIKROGEN and LAISON were compared.

  1. Capai L, Falchi A, Charrel R: Meta-Analysis of Human IgG anti-HEV Seroprevalence in Industrialized Countries and a 389 Review of Literature. Viruses 2019, 11(1). 390

Response 2: This article is a seroprevalence study and only tested HEV-IgG. 

  1. Shrestha AC, Flower RL, Seed CR, Stramer SL, Faddy HM: A Comparative Study of Assay Performance of Commercial 391 Hepatitis E Virus Enzyme-Linked Immunosorbent Assay Kits in Australian Blood Donor Samples. J Blood Transfus 2016, 392 2016:9647675.

Response 2: The authors tested samples in Australian blood donor (n=394). 194 samples were positive for HEV-IgG and of these only 4 samples were HEV-IgM positive. In this article neither checked convalescence phase.

Several other publications are available regarding this.

Comment 3: Thus, any new article should contain novelty and new information regarding this issue. There may be several reasons for slight differences in sensitivity and specificity between different kits based on the time of collection of sera, patients’ condition, and others.

Another area of interest is to provide a response about the utility of different kits depending on work with a multicenter character. The utility and sensitivity of the kit may be variable among different populations worldwide.

Response 3: Different published articles showed discrepancies in sensitivity and specificity but most of them only tested HEV-IgG or HEV-IgM and compared less techniques. In our article the authors include four commonly used techniques in European countries with the same samples and include convalescence phase. Most of the available techniques show discrepancies in convalescence phase and is important to clarify.

Reviewer 3 Report

Comments and Suggestions for Authors

This manuscript describes four different antibodies against Hepatitis E virus. HEV induces zoonotic disease and considering as important pathogen.

Story looks too simple but still have important message from findings.

Here are comments and suggestion.

Better to change as Brief Report.

As story looks simple but still need detailed information and description required on all Tables.

Describes explanation and conclusion right after each table.

Table 1 and results, table 2 and results and meaning. Table 3 followed by description and meaning so that readers understand what is the meaning and conclusion.

Thanks

Comments on the Quality of English Language

fine

Author Response

Reviewer 3

Comment 1: This manuscript describes four different antibodies against Hepatitis E virus. HEV induces zoonotic disease and considering as important pathogen.

Story looks too simple but still have important message from findings.

Here are comments and suggestion.

Better to change as Brief Report.

Response 1: Due to the controversy in the results using different diagnostic texts, the authors consider that the article is relevant since it includes more techniques than other authors and with a greater number of samples. In addition, it includes the convalescence phase, which is the one in which most techniques have discordant results. It is not possible to condense all results in a brief report.

Comment 2: As story looks simple but still need detailed information and description required on all Tables.

Response 2: The authors have included supplementary material with raw data for readers have more detailed information.

Comment 3: Describes explanation and conclusion right after each table.

Table 1 and results, table 2 and results and meaning. Table 3 followed by description and meaning so that readers understand what is the meaning and conclusion.

Response 3: The authors have changed the order of the tables and the meaning and conclusion. First include the explanation and after the table for greater understanding of readers.

Thanks

Comments on the Quality of English Language: fine

Round 2

Reviewer 1 Report

Comments and Suggestions for Authors

The authors have adequately addressed my concerns in their revised manuscript. I read through the manuscript again and only have a few minor comments:

1) Line 11: ...HEV infection...

2) Line 31: ...subfamily Orthohepevirinae...

3) Line 36: HEV-3

4) Line 107: HEV-1 to HEV-4

5) Section 2.4 and lines 183-185: Why did the authors not include a phylogenetic tree in this manuscript since you have conducted a phylogenetic analysis?

6) Table 3: ...0.394, 0.001, 0.019, 0.001...

Author Response

Reviewer 1

The authors have adequately addressed my concerns in their revised manuscript. I read through the manuscript again and only have a few minor comments:

Comment 1: Line 11: ...HEV infection...

Response 1: The authors have corrected it.

Comment 2: Line 31: ...subfamily ..

Response 2: The authors have corrected it.

Comment 3: Line 36: HEV-3

Response 3: The authors have corrected it.

Comment 4: Line 107: HEV-1 to HEV-4

Response 4: The authors have corrected it.

Comment 5: Section 2.4 and lines 183-185: Why did the authors not include a phylogenetic tree in this manuscript since you have conducted a phylogenetic analysis?

Response 5: The authors have included phylogenetic tree in supplementary material.

6) Comment 6: Table 3: ...0.394, 0.001, 0.019, 0.001...

Response 6: The authors have corrected it.

Reviewer 2 Report

Comments and Suggestions for Authors

A revised version of the article has been submitted. It has been checked by analyzing various aspects of the article. When the original version of the article was reviewed, the significant limitations of the article were focused. These comments mainly exposed the limitations of the article regarding:

1.      Novelty

2.      Lack of new information

3.      Similarity with other articles and focusing on similar information

4.      Utility of different kits and their limitations

5.      Clinical implications of the article

In the revised version, the authors have not responded to these queries and comments by providing additional experiments. Also, they could not explain the reviewer's queries. No evidence was submitted to support their article and the queries. Instead, they have responded without any evidence or new information.

Author Response

A revised version of the article has been submitted. It has been checked by analyzing various aspects of the article. When the original version of the article was reviewed, the significant limitations of the article were focused. These comments mainly exposed the limitations of the article regarding:

  1. Novelty

Response 1. The authors really think that novelty is included in the article as there is not any other article in which sensitivity is compared in the four chosen techniques: LIAISON® MUREX anti-HEV IgG and anti-HEV IgM, Hepatitis E VIRCLIA® IgM and IgG monotests, Wantai HEV IgM and IgG ELISA and VIDAS® anti-HEV IgM and IgG tests. Besides, we distinguish the positive IgM samples in acute and convalescence phase, which is not done in most articles. That is why, our correlation agreement is not as higher as in other studies, what must be considered if techniques are interchangeable.

  1. Lack of new information

Response 2. Our article stated that Wantai HEV IgM and IgG ELISA was not the technique with the best results in sensitivity comparing with the other techniques previously mentioned, especially in convalescence phase samples (lines 233-238). Besides, new information is obtained from Cohen's kappa coefficient as it has been commented before (lines 253-263). Furthermore, according to IgG results our study shows that VIRCLIA showed certain number of false positive results and this information is essential to seroprevalence studies.

  1. Similarity with other articles and focusing on similar information

Response 3. Articles comparing sensitivity in different diagnosis techniques are focused on similar information. Nevertheless, our article includes more techniques than other authors and with a greater number of samples. In addition, it includes the convalescence phase, which is the one in which most techniques have discordant results.

  1. Utility of different kits and their limitations

The authors comment this point throughout the discussion.

  1. Clinical implications of the article

Response 5. Hepatitis E infections are increasing each year. That is why, it is needed high throughput assays for quicker analysis and, as our article shows, LIAISON® MUREX anti-HEV IgM can be considered the first one based on chemiluminescence to resolve infections. On the other hand, LIAISON® MUREX anti-HEV IgG can be used in seroprevalence studies as it can work with a huge quantity of samples in a short period of time (30 minutes per sample). ). The authors comment this argument in discussion. 

In the revised version, the authors have not responded to these queries and comments by providing additional experiments. Also, they could not explain the reviewer's queries. No evidence was submitted to support their article and the queries. Instead, they have responded without any evidence or new information.

Reviewer 3 Report

Comments and Suggestions for Authors

Much better now and accept.

Author Response

Thank you